# Protective effect of propofol compared with sevoflurane on liver function after hepatectomy with Pringle maneuver: A randomized clinical trial

**Junya Matsumi** 📧 *, **Tetsufumi Sato**

Department of Anesthesia and Intensive Care, National Cancer Center Hospital, Tokyo, Japan

* jmatsumi@ncc.go.jp

## Abstract

While the Pringle maneuver reduces intraoperative blood loss in hepatectomies, this technique can also be hepatotoxic. Hepatectomies require general anesthesia with propofol or volatile anesthetics like sevoflurane, agents known to offer multi-organ protection. However, their clinical effect after liver resection is unclear. We aimed to assess the effect of the two anesthetics on post-hepatectomy liver damage via measuring liver function tests. Fifty-six patients who underwent elective hepatectomies with the Pringle maneuver due to metastatic hepatic masses were preoperatively randomized to be anesthetized by sevoflurane or propofol. The primary and secondary outcomes were the postoperative peak levels of aspartate transaminase (AST) and alanine transaminase (ALT), respectively. Patients anesthetized by propofol exhibited significantly lower transaminases than those given sevoflurane (AST, p = 0.005; ALT, p = 0.006). The former agent significantly affected postoperative transaminases (AST hazard ratio -192.2, 95% confidence interval [-332.1 to -52.4], p = 0.00; ALT hazard ratio -140.2, 95% confidence interval [-240.0 to -40.7], p = 0.007). In conclusion, propofol had a greater hepatoprotective effect than sevoflurane as assessed by postoperative transaminases after hepatectomy with Pringle maneuver for metastatic liver tumors.

**Data Availability Statement:** All relevant data are within the manuscript.

## Introduction

A hepatectomy is crucial in treating various primary and secondary liver tumors. Despite the improvement in perioperative management and surgical technique, the mortality and morbidity rates after hepatectomy remain around 2–4% and 20–45%, respectively [1–4]. A main problem during liver resection is hemorrhage [5, 6]. Inflow occlusion by clamping the hepatoduodenal ligament, or the Pringle maneuver, is commonly done to reduce intraoperative blood loss [7]. However, this technique also induces liver damage.

Currently, there are protective strategies to prevent damage during a hepatectomy [8]. One established technique is ischemic preconditioning with intermittent Pringle maneuver [9]. Another is pharmacologic therapy. Some potentially effective drugs include propofol and

**Funding:** This study was supported by The National Cancer Center Research and Development Fund (29-A-12). The funders had no role in study design, data collection and analysis, decision to publish, or preparation of the manuscript.

**Competing interests:** The authors have declared that no competing interests exist.

volatile anesthetic agents, commonly used to maintain general anesthesia. These drugs have been shown to be hepatoprotective in experimental studies [10–12].

However, research on the two drugs has been conflicting. Some studies have demonstrated that propofol is more protective than volatile anesthetic agents during hepatectomies in humans [13, 14]. In contrast, there is also literature on the use of sevoflurane resulting in lower transaminase levels during a hepatectomy using the Pringle maneuver [15, 16]. Meanwhile, other studies show no difference between the effects of sevoflurane and propofol on postoperative transaminase levels [17, 18]. Moreover, some studies had different methodologies, such as pharmacologic postconditioning and ischemic preconditioning, from the usual clinical setting [15, 16]. Also, patients with cirrhosis were included in some studies, and the size of excised liver was also considered [15–18]; factors other than anesthetic agents might influence transaminases. Thus, there is a need to study the hepatoprotective effect of propofol versus sevoflurane in the typical clinical and more homogeneous setting.

This study aimed to assess the hypothesis that propofol-based anesthesia in patients without cirrhosis is more protective than sevoflurane-based anesthesia against liver damage as evaluated via liver transaminases during minor hepatectomy with the Pringle maneuver.

## Materials and methods

### Study design and settings

This randomized controlled trial was conducted at National Cancer Center Hospital (NCCH) in Tokyo, Japan. It was approved by the institutional review board for human clinical studies (no. 2017–504), and written informed consent was obtained from all patients. The study was carried out following the Declaration of Helsinki and registered in the University Hospital Medical Information Network (UMIN) Clinical Trial Registry (UMIN000034798) on December 12, 2018.

### Inclusion and exclusion criteria

Between January 11, 2019, and November 19, 2020, we included patients diagnosed with metastatic or suspected metastatic hepatic masses undergoing elective liver resection with the Pringle maneuver at NCCH.

Exclusion criteria included patients aged more than 90 or under 18, a diagnosis of liver cirrhosis (LC), preoperative liver transaminases over 100 IU/L, more than five preoperatively diagnosed metastatic hepatic tumors, scheduled hepatic lesion resection in more than five Couinaud segments, concomitant additional therapy (such as the resection of the primary lesion and radiofrequency ablation) or biliary duct reconstruction, known allergies to the trial anesthetics, and patient refusal or withdrawal.

### Randomization

Enrolled patients were preoperatively randomized to be anesthetized by either sevoflurane or propofol. Blocked randomization without stratification was performed by one of the authors not involved in informed consent acquisition and anesthetic administration.

### Perioperative management

According to the predefined NCCH clinical protocol, all patients received similar perioperative treatment except for those who experienced postoperative complications. None received premedication for anesthesia. Continuous ropivacaine infusion via the thoracic epidural route

was administered for postoperative pain management. In cases without epidural anesthesia or when the effect of epidural anesthesia was inadequate, fentanyl was given intravenously.

All patients underwent radial arterial invasive blood pressure monitoring. General anesthesia, such as propofol, rocuronium, fentanyl, and remifentanil, was used for induction. After induction, propofol was infused by titrating for bispectral index (BIS) values between 30 to 70 in the propofol group. Sevoflurane was administered by titrating the end-tidal concentration of sevoflurane between 0.6–2% in the sevoflurane group. There were no limitations in perioperative management without using an anesthetic agent not assigned.

## Surgical procedure

One of five surgeons certified in hepatobiliary surgery performed the hepatic resections in a standardized manner. The Pringle maneuver was intermittently performed (cycles of 15 to 30 min of ischemia followed by 5 min of reperfusion) using a large vascular clamp with rubber jaws or a vascular clip [9]. The forceps clamp-crush or cavitron ultrasonic surgical aspirator method was used for parenchymal transection. Further, the exposed vessels were ligated with silk threads or sealed with surgical instruments.

## Outcomes

The primary outcome was the peak level of aspartate aminotransaminase (AST) during three postoperative days (POD), representing postoperative liver injury.

The secondary outcomes, measured during three POD, were peak alanine aminotransaminase (ALT) levels to measure liver injury and total bilirubin (tBil) to measure liver function. Another secondary outcome included postoperative serious adverse events (SAE) occurring within 28 POD. SAE was defined as Grades 3a, 3b, 4a, 4b, and 5 according to the Common Terminology Criteria for Adverse Events, version 4.0, of the Japan Clinical Oncology Group. A previous study at NCCH confirmed that AST, ALT, and tBil peaked within three POD [9].

## Measurements

We prospectively collected perioperative parameters. The parameters collected were age, sex, American Society of Anesthesiologists Physical Status (ASA-PS), Charlson Comorbidity Index, LC and hepatic steatosis histopathologically proven by postoperative liver samples, treatment with chemotherapy within one year, the primary malignancy, duration of surgery (time between skin incision to closure), anesthesia time (time between the start of anesthesia induction to the patient leaving the operation room), the sum of the time in the Pringle maneuver during liver resection (total ischemic time [TIT]), intraoperative fluid balance, and weight of the resected liver.

## Statistical analyses

We summarized the assessed variables using the mean and standard deviation or percentage (%) using Student's t-test or Fisher's exact test where appropriate. A multivariable linear regression using age, gender, TIT, and the anesthetic agent used was utilized to determine factors that affected postoperative peak levels of AST and ALT. Confidence intervals (CI) of hazard ratios (HR) were estimated at 95%. All outcomes were analyzed according to the intention-to-treat analysis.

Additionally, as a sub-analysis, all outcomes were analyzed according to the per-protocol effect. A p-value less than 0.05 was considered to be statistically significant.

All analyses were performed with EZR (Saitama Medical Center, Jichi Medical University, Saitama, Japan), a graphical user interface for R (The R Foundation for Statistical Computing, Vienna, Austria) [19].

### Sample size estimation

We preliminarily collected the data from 52 patients who underwent hepatectomy for metastatic masses at NCCH: the postoperative peak of AST was 658 [460] IU/L with sevoflurane and 351 [178] IU/L with propofol (S1 Table). As actual differences were 100 to 300 IU/L in previous studies and retrospective data at NCCH, and the standard deviation was 200 to 400 IU/L after hepatectomy, we set a difference of 300 IU/L in postoperative peak AST levels between the two groups as clinically significant. The standard deviation was determined to be 400 IU/L in postoperative peak AST levels [15–18]. Then, the sample size was calculated based on a difference of 300 IU/L in postoperative peak AST levels between the two groups and a standard deviation of 400 IU/L in postoperative peak AST levels with a type 1 error of 0.05, and a power of 0.8. This analysis showed that 28 patients were required in each group (56 patients).

## Results

### Patients

Fifty-six patients were included between January 11, 2019, and November 19, 2020. During this period, 117 patients underwent hepatectomies for metastatic tumors at NCCH. Fifty-three patients did not meet the inclusion criteria (37 patients had scheduled resection of other organs, 13 were preoperatively diagnosed with six or more tumors, and three had liver lesions involving more than five Couinaud segments). Eight patients refused to participate in this study (Fig 1). There were deviations from the study protocol in three patients: one patient in the sevoflurane group received another anesthetic agent perioperatively, and the Pringle maneuver could not be performed in one patient in each group due to adhesions from a previous surgery.

### Characteristics

Table 1 shows the patients' characteristics. Colorectal cancer was the most frequent primary lesion. No LC was diagnosed histopathologically in the two groups. Anesthesia and operation times were long in the sevoflurane group. Additionally, none of the patients in this study underwent intraoperative revascularization procedures, such as portal vein or hepatic artery reconstruction, during liver resection. Also, none received continuous perioperative administration of vasopressors.

### Outcomes

The outcomes differed significantly between the two groups (Table 2). Both AST and ALT levels presented as mean (standard deviation), were significantly lower in the propofol group (AST: sevoflurane 510.3 [362.4] IU/L versus propofol 291.7 [151.2] IU/L; p = 0.005 and ALT: sevoflurane 422.1 [257.3] IU/L; propofol 264.3 [131.4] IU/L; p = 0.006).

The secondary outcomes were not significantly different between the two groups (tBil: sevoflurane 1.24 [0.53] mg/dL versus propofol 1.16 [0.41] mg/dL; p = 0.55. SAE: sevoflurane 10.7% versus propofol 7.1%; p = 1). SAE included four patients who developed bile fistula or intra-abdominal abscesses needing drainage (two patients per group) and one with urosepsis (in the sevoflurane group).

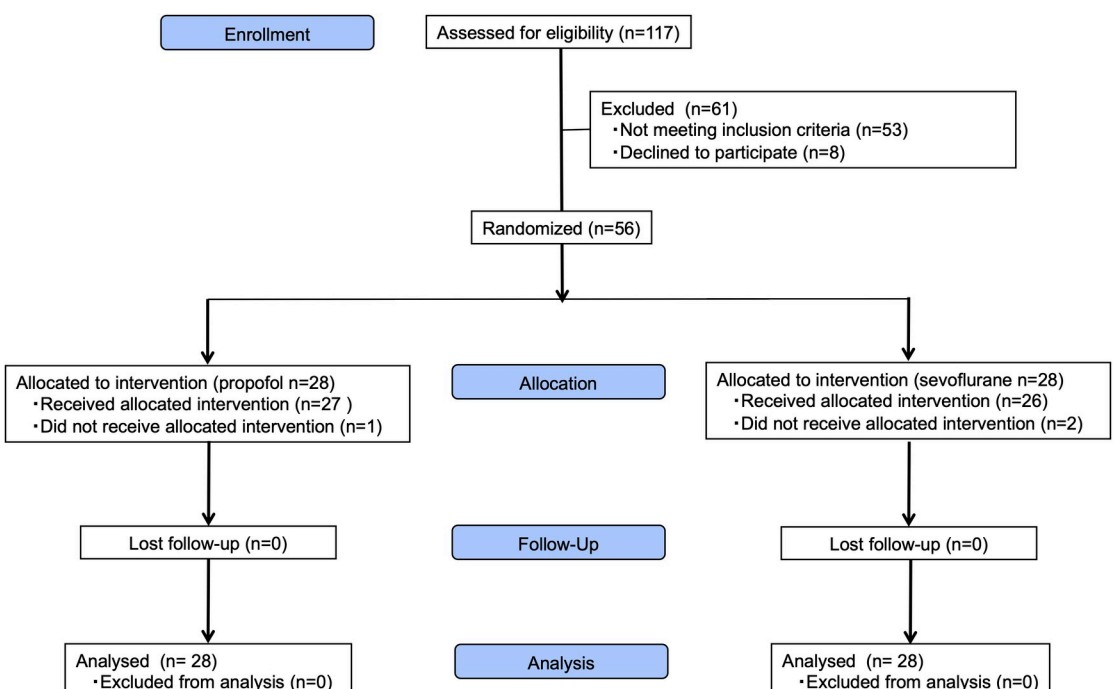

**Fig 1. Enrollment and randomization.** Sixty-one patients were excluded due to scheduled resection of other organs. Reasons for exclusion included being diagnosed with six or more tumors, liver lesions involving more than five Couinaud segments, and refusal to participate in the study.

Postoperative peak values of AST and ALT were significantly affected by the choice of anesthetic agent (propofol p = 0.008 and 0.007, respectively) and TIT (p = 0.001 and p = 0.0004, respectively, Table 3).

## Per-protocol analysis

The per-protocol analysis included 26 and 27 patients in the sevoflurane and propofol groups, respectively. The characteristics were similar to the original analysis (S2 Table), whereas the primary and secondary outcomes were the same as the original (S3 and S4 Tables)

## Discussion

In this randomized control trial, we noted significantly lower postoperative peak transaminase levels in patients anesthetized by propofol than in those given sevoflurane after metastatic liver tumor resection with the Pringle maneuver.

Several studies have explored the protective effect of propofol against organ damage, including liver injury, in experimental models [20–24]. Additionally, some studies have shown that propofol has a more substantial protective effect than volatile anesthetics during human liver resection, as assessed by biomarkers not commonly used in a clinical setting [13, 14]. Based on these results, transaminases may be used to assess the protective effect of anesthetic agents against liver damage. However, our results are inconsistent with other studies' findings.

A series of studies by Beck-Schimner and colleagues showed the beneficial effect of sevoflurane against liver damage after liver resection using the Pringle maneuver [15, 16]. The patients were mainly anesthetized with propofol, and sevoflurane was used only briefly before or after inducing ischemia. These anesthetic methods are not usually performed in a clinical

**Table 1. Patients' characteristics.**

|  |  | Sevoflurane n = 28 | Propofol n = 28 |
|---|---|---|---|
| Age (years) |  | 66.3 [12.3]* | 64.7 [10.1] |
| Male (%) |  | 67.9 | 71.4 |
| ASA-PS (%) | 2 | 85.7 | 89.3 |
|  | 3 | 14.3 | 10.7 |
| Charlson Comorbidity Index |  | 6.5 [0.7] | 6.6 [0.8] |
| Fatty liver (%) |  | 14.3 | 7.1 |
| Chemotherapy within 1 year (%) |  | 28.6 | 25.0 |
| Primary cancer site (%) | Colorectal | 75.0 | 78.6 |
|  | Others | 25.0 | 21.4 |
| Preoperative AST level (IU/l) |  | 27.4 [12.3] | 26.4 [12.3] |
| Preoperative ALT level (IU/l) |  | 24.9 [19.9] | 24.0 [13.7] |
| Preoperative tBil level (mg/dl) |  | 0.8 [0.4] | 0.8 [0.2] |
| Laparoscopic surgery (%) |  | 17.9 | 17.9 |
| Use of epidural anesthesia (%) |  | 96.4 | 100 |
| Operation time (min) |  | 290.4 [98.2] | 237.4 [46.6] |
| Anesthesia time (min) |  | 352.0 [106.1] | 305.1 [54.1] |
| Total ischemic time (min) |  | 68.5 [42.7] | 59.7 [27.0] |
| Number of Pringle cycles (cycle) |  | 4.3 [3.0] | 3.5 [1.8] |
| Intraoperative bleeding (ml) |  | 607.6 [594.0] | 446.4 [370.7] |
| Intraoperative fluid balance (ml/kg/h) |  | 7.1 [3.3] | 6.8 [2.1] |
| Weight of resected liver (g) |  | 121.5 [97.4] | 158.9 [129.8] |

ASA-PS: American Society of Anesthesiologists Physical Status, AST: aspartate aminotransaminase, ALT: alanine aminotransaminase, tBil: total bilirubin.

* Data are presented as the % of the total number or mean [standard deviation].

setting. In turn, the anesthetic methods utilized in our study were the standard of care. One study included other ischemic methods that can affect postoperative transaminase levels [16]. Consequently, the effects seen in these studies might not reflect the impact of anesthetic agents on postoperative liver damage in the usual clinical setting.

Other studies by Song and Slankamenac showed no difference between sevoflurane and propofol [17, 18]. In contrast, this study did not include patients with LC. We thus speculate that these previous studies could not compare the effects of anesthetic agents against liver damage during hepatectomy with the Pringle maneuver without significant bias.

**Table 2. Outcome parameters at univariable analysis.**

|  | Sevoflurane n = 28 | Propofol n = 28 | p-value** |
|---|---|---|---|
| Peak AST level (IU/l) | 510.3 [362.4]* | 291.7 [151.2] | 0.005 |
| Peak ALT level (IU/l) | 422.1 [257.3] | 264.3 [131.4] | 0.006 |
| Peak tBil level (mg/dl) | 1.24 [0.53] | 1.16 [0.41] | 0.55 |
| SAE (%) | 10.7 | 7.1 | 1 |

AST: aspartate aminotransaminase, ALT: alanine aminotransaminase, tBil: total bilirubin, SAE: serious adverse events.

* Data are presented as the % of the total number or mean [standard deviation].

** A p-value less than 0.05 was considered to be statistically significant.

**Table 3. Multivariable linear regression of postoperative peak transaminase levels.**

| Factor | AST | | | ALT | | |
|---|---|---|---|---|---|---|
| | Hazard ratio | 95%CI | P | Hazard ratio | 95%CI | P value* |
| Age | -0.9 | -7.5 to 5.6 | 0.78 | -1.9 | -6.6 to 2.7 | 0.41 |
| Female | 39.3 | -117.2 to 195.7 | 0.62 | 65.3 | -46.0 to 176.6 | 0.24 |
| TIT | 3.3 | 1.4 to 5.3 | 0.001 | 2.6 | 1.2 to 4.0 | 0.0004 |
| Propofol | -192.2 | -332.1 to -52.4 | 0.008 | -140.2 | -240.0 to -40.7 | 0.007 |

AST: aspartate aminotransaminase, ALT: alanine aminotransaminase, TIT: total ischemic time, CI: confidence interval.

*A p-value less than 0.05 was considered to be statistically significant.

This study set careful selection criteria for its participants to avoid potential biases. As the size of the resected liver may affect postoperative transaminase levels, we only selected patients with a set number of metastatic liver tumors and liver lesions involving less than five Couinaud segments. Consequently, no significant difference in the size of the resected liver (assessed by resected liver weight) was noted in this study.

Second, TIT may also affect postoperative transaminase levels. Since we determined that the resected liver size was similar among the study participants, we speculated that the TIT during metastatic liver tumor resection would be similar. Further, the results revealed that the TIT was similar between the two groups.

Third, baseline liver function can also affect postoperative transaminase levels [24]. Since many patients with HCC also have varying degrees of LC, the liver function of patients with HCC varies widely. In contrast, patients with metastatic hepatic tumors may have similar liver function. As a result, none of our patients had LC, and the frequency of hepatic steatosis between the two groups was similar.

Overall, we think the homogenization of baseline characteristics that could affect the primary outcome was achieved. Thus, the comparison between the protective effect of anesthetic agents against liver damage could be elucidated.

This study has several limitations which should be considered when interpreting its results. First, as we usually use end-tidal sevoflurane concentration monitoring during sevoflurane anesthesia, BIS was not used for titration in the sevoflurane group. Nevertheless, end-tidal anesthetic concentration monitoring is as reliable as BIS; hence, the depth of anesthesia was compatible between groups [25]. Second, a large standard deviation was set at sample size estimation, which could lead to decreased power. However, the value of postoperative transaminase also varies widely in several studies [15–18], and this study's achieved power was not very low (a power of 78.8%, calculated by the difference in mean values between groups 218.6 IU/L [SD 296.4 IU/L] and a type 1 error 0.05).

Third, in the sevoflurane group, anesthesia and operation times were long. The difference between the former and the latter was similar;and operation time was more influential in the difference noted. As the TIT, representing the duration of liver resection, was similar between the groups, the duration of liver resection and the influence of liver ischemia might similar. These times might concern with Adhesive detachment for previous surgeries including resection of the primary malignancy. Hence, the difference in these periods might not largely affect our results.

Fourth, we did not perform protocolized intraoperative management, such as maintaining a low central venous pressure. Such practices may affect intraoperative fluid volume and blood loss. However, the intraoperative fluid balance and blood loss differed between the two groups. Thus, this matter might not be significantly influential. Fifth, there was a deviation from the study protocol in three patients due to clinically justifiable causes. Nonetheless, as the predefined per-protocol analysis showed similar results, its effect was considered minimal.

Lastly, SAE incidence, the most critical measure for patients, was similar between groups in this study. Although this study aimed to assess anesthetic agents' effects on liver damage during liver resection with the Pringle maneuver, we included only minor hepatectomies to maintain baseline characteristics' homogenization. Therefore, the incidence of SAE in the patients included in this study was essentially low. Thus, the statistical power for evaluating the influence on SAE is very weak. Further research is needed to assess whether propofol-based anesthesia lowers the incidence of SAE compared with sevoflurane-based anesthesia after major liver resection with the Pringle maneuver.

In conclusion, our randomized controlled study indicates that propofol may have a more substantial protective effect during liver resection with the Pringle maneuver in a clinical setting.

## Supporting information

**S1 Checklist. CONSORT checklist.**
(DOC)

**S1 Table. The preliminary study's data set.** This table contains the data set for the power calculation.
(XLSX)

**S2 Table. Patients' characteristics in a per-protocol analysis.** ASA-PS: American Society of Anesthesiologists Physical Status, AST: aspartate aminotransaminase, ALT: alanine aminotransaminase, tBil: total bilirubin. Data are presented as the % of the total number or mean [standard deviation], where appropriate.
(DOCX)

**S3 Table. Outcome parameters at univariable analysis in a per-protocol analysis.** AST: aspartate aminotransaminase, ALT: alanine aminotransaminase, tBil: total bilirubin, SAE: serious adverse events. Data are presented as the % of the total number or mean [standard deviation], where appropriate. A p-value less than 0.05 was considered to be statistically significant.
(DOCX)

**S4 Table. Multivariable linear regression of postoperative peak transaminase levels in a per-protocol analysis.** AST: aspartate aminotransaminase, ALT: alanine aminotransaminase, TIT: total ischemic time, 95%CI: 95% confidence interval. A p-value less than 0.05 was considered to be statistically significant.
(DOCX)

**S5 Table. The study's minimal underlying data set.** This table contains the data set underlying the results described in the manuscript.
(CSV)

**S1 File. Study protocol (Japanese).**
(DOCX)

**S2 File. Study protocol (English).**
(DOCX)

## Acknowledgments

I wish to thank Dr. Kazuaki Shimada, Dr. Minoru Esaki, Dr. Satoshi Nara, Dr. Daisuke Ban, and Dr. Yoji Kishi for advice regarding the surgical technique for hepatectomy.

## Author Contributions

**Conceptualization:** Junya Matsumi.

**Data curation:** Junya Matsumi.

**Formal analysis:** Junya Matsumi.

**Funding acquisition:** Tetsufumi Sato.

**Investigation:** Junya Matsumi.

**Methodology:** Junya Matsumi.

**Project administration:** Junya Matsumi, Tetsufumi Sato.

**Supervision:** Tetsufumi Sato.

**Writing – original draft:** Junya Matsumi.

**Writing – review & editing:** Junya Matsumi, Tetsufumi Sato.

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
