## [Decision Letter · Decision Letter 0]

13 Jun 2023

PONE-D-23-10418

Protective effect of propofol compared with sevoflurane on liver damage during liver resection with Pringle maneuver: a randomized clinical trial

PLOS ONE

Dear Dr. Matsumi,

Thank you for submitting your manuscript to PLOS ONE. After careful consideration, we feel that it has merit but does not fully meet PLOS ONE’s publication criteria as it currently stands. Therefore, we invite you to submit a revised version of the manuscript that addresses the points raised during the review process.

ACADEMIC EDITOR: 

Kindly review my points that need to be corrected in the manuscript.

Address all the points highlighted by the Reviewers.

Please submit your revised manuscript by Jul 28 2023 11:59PM If you will need more time than this to complete your revisions, please reply to this message or contact the journal office at plosone@plos.org. Please include the following items when submitting your revised manuscript:

We look forward to receiving your revised manuscript.

Kind regards,

Hossam Eldien Ahmed Anis ElShamaa, M.D.

Academic Editor

PLOS ONE

Journal Requirements:

2. Please upload a new copy of Figure 1 as the detail is not clear. Please follow the link for more information: " ext-link-type="uri" xlink:type="simple">https://blogs.plos.org/plos/2019/06/looking-good-tips-for-creating-your-plos-figures-graphics/"
https://blogs.plos.org/plos/2019/06/looking-good-tips-for-creating-your-plos-figures-graphics/"

Additional Editor Comments:

Thank you for this well done manuscript, but I have minor points that need to be addressed;

1. Methodology: in perioperative management , titration of propofol and sevoflurane should be based both on the BIS value to have solid comparison based on np differences.

2. in the manuscript the author have to clarify both ( Anesthesia time surgery time ) , it should be stated from exactly when to when.

3. All tables should have a caption clearly stating the type of date and how is it expressed and the P value of statistical significance.

4. Figure 1 has neither title nor caption.

5. In the pre-protocol the author showed the operative time to be significantly longer in the Sevoflurane group,,, why ?

6. Discussion: paragraph 1, in the last line , the author said (anesthetic agent was significantly affected with postoperative peak levels of transaminase.) what is meant by this phrase?

Reviewers' comments:

Reviewer's Responses to Questions

**Comments to the Author**

1. Is the manuscript technically sound, and do the data support the conclusions?

Reviewer #1: Yes

Reviewer #2: Yes

2. Has the statistical analysis been performed appropriately and rigorously? 

Reviewer #1: Yes

Reviewer #2: I Don't Know

3. Have the authors made all data underlying the findings in their manuscript fully available?

Reviewer #1: Yes

Reviewer #2: Yes

4. Is the manuscript presented in an intelligible fashion and written in standard English?

Reviewer #1: Yes

Reviewer #2: No

5. Review Comments to the Author

Reviewer #1: Important note: This review pertains only to ‘statistical aspects’ of the study and so ‘clinical aspects’ [like medical importance, relevance of the study, ‘clinical significance and implication(s)’ of the whole study, etc.] are to be evaluated [should be assessed] separately/independently. Further please note that any ‘statistical review’ is generally done under the assumption that (such) study specific methodological [as well as execution] issues are perfectly taken care of by the investigator(s). This review is not an exception to that and so does not cover clinical aspects {however, seldom comments are made only if those issues are intimately / scientifically related intermingle with ‘statistical aspects’ of the study}. Agreed that ‘statistical methods’ are used as just tools here, however, they are vital part of methodology [and so should be given due importance]. I look at the manuscript in/with statistical view point, other reviewer(s) look(s) at it with different angle so that in totality the review is very comprehensive. However, there should be efforts from authors side to improve (may be by taking clues from reviewer’s comments). Therefore, please do not limit the revision only (with respect) to comments made here.

COMMENTS: Although it is observed that this manuscript is well drafted [and the study is excellent with respect to most of the aspects], I have few observations/concerns (different opinion) which are given below:

I have few (minor/little) doubts regarding ‘Sample size estimation’. Firstly, you said “We set a difference of 300 IU/L in postoperative peak AST levels between the two groups” but the basis of this is not given. Though later as you specified since it is felt to be ‘clinically significant’ is perfectly alright but mind you that unless any reference is quoted, it is treated as ‘subjective’ [I am not a clinician, however, general opinion is expressed]. For standard deviation you said “was decided by referring to retrospective data of 52 patients” but the source of these data is not revealed. At the end of this section you said “this study was adequately powered with 28 patients per group (total 56 patients)” is doubtful because according to table-2 on page 158 of Jacob Cohen’s paper “A power primer” in Psychological Bulletin, 1992, vol.:112, pp 155-159 [which is a sort of summary of the excellent book by Cohen himself titled ‘Statistical power analysis for the behavioral sciences’, Academic Press, 1977, New York] even for medium effect size you need n=64 per group (type-I error=0.05, power=80%). Fortunately, the power achieved in this trial is not very low (75.69% by referring to table-2 of this manuscript), however, here the comparison pertains only to ‘post’ values and baseline is not given any recognition [I am not sure regarding its clinical importance/relevance]. Look at the absolute values [both mead S.D.] in table-2 {Peak AST level (IU/l) Sevoflurane group (n=28) 510.3 [362.4] Propofol group (n=28) }291.7 [151.2]}.

Any software or manual calculations will show that your ‘sample size’ results are definitely correct [as an example, a ‘Screen-shot’ of output from software called COMPARE2 is pasted below – which shows that I have no doubt about correctness of calculations], I doubt only about assumptions [assumed values of a difference of 300 IU/L and a standard deviation of 400 IU/L]. This implies that a very large effect size is assumed and that is doubtful.

Example: Screen-shot of output from COMPARE2

Note that the comparison of baseline characteristics when random allocation/assignment is used/done is not required [‘P’ values in last column 4 of Table 1 - Patients characteristics]. In this context, please read [though I am sure that the authors already know these things] a note which is pasted from one famous standard textbook on ‘Medical Research Methodology’:

To provide a description of baseline characteristics is entirely reasonable (since it is clearly important in assessing to whom the results of the trial can be applied), however, statistical comparison of baseline characteristics when random allocation/assignment is used/done [often for good/standard/leading journals these days] is not required, because even if P-value(s) turn(s) out to be significant (while comparing baseline characteristics despite random allocation), it is, by definition, a false positive as you then are supposed to be testing ‘randomization’ then, which in any single trial may not balance all baseline characteristics (particularly when sample sizes are small). Remember that ‘randomization’ is a sort of ‘insurance’ and not a guarantee scheme. Authors may please refer to following articles:

References:

1. Stuart J. Pocock, et al., ‘Subgroup analysis, covariate adjustment and baseline comparisons in clinical trial reporting: current practice and problems’, Statistics in medicine, 2002; 21:2917–2930 [Particularly page 2927]

2. Harrington D, et al., ‘New guidelines for statistical reporting in the journal’, N Engl J Med 2019;381:285-6

[Important message (indirectly/ultimately indicated) from these articles: Never do any comparison with respect to ‘baseline’ characteristics {by applying statistical significance test(s)}, when allocation is done randomly].

However, Statistical comparison [only with respect to important/indicated variables] of baseline characteristics may be performed, to find out if analysis adjustment (say stratified analyses or else) is required with respect to these variables.

In ‘Abstract- Results’ section [and also in ‘outcome - Per-protocol analysis’ section later] there is a sentence “At multiple linear regression …..” which, in my opinion, is not grammatically correct {instead of At, it should be As shown by} . Please check. Later in ‘Randomization’ section, a sentence “A computer generated the blocked randomization with no stratification” appears to be incomplete. In section ‘Statistical analyses’, instead of “We expressed the assessed variables….” it could/should [desired] be “We summarised the assessed variables….”, I think.

Highlighting (listing/pointing out) the “Limitations” {with possible effects} is highly appreciable (although lower sample size is not included starting ‘Discussion’ with “In this adequately powered RCT is not agreed). However, mind you that as pointed out in ‘important note’ above “This review pertains only to ‘statistical aspects’ of the study and so ‘clinical aspects’ should be assessed separately/independently [preferably by an expert surgeon in concerned field]. Some of these limitations may have significant influence on results.

Except these minor points, the article is acceptable. ‘Minor Revision’ is recommended.

Reviewer #2: In Manuscript PONE -D-23010418, Matsumi and Sato present their recent research examining the protective effect of propofol (P) vs sevoflurane (S) on liver damage during liver resection with a Pringle maneuver. This work was in part a further examination of these anesthetic agents on liver injury, which had previously been examined with conflicting results. I commend the authors for attempting to perform a more thorough randomized study to address this question. The collected data and results are presented in a similar to prior work, so direct comparisons are possible. Overall, the results are compelling however, the paper needs English editing as well as some stylistic editing as detailed below.

1. The Introduction discusses prior research in this area but should also include a better description of the differences in protocols between the prior study (See Discussion Page 22 “The patients were mainly anesthetized in…. usual clinical methods”). By including this discussion in the Introduction would better inform the reader about why this study is important. The authors used a protocol that more closely mimicked traditional anesthetic practice compared to prior work by others. This point should be emphasized more.

2. The authors excluded patients with liver cirrhosis, only included patients with similar tumor burdens etc. to produce a more homogenous patient population (see last paragraph in the Discussion). Such details should be emphasized in the introduction to build interest in the reader.

3. Propofol was titrated based upon bispectral index monitoring, but sevoflurane was titrated to end-tidal concentration. Why was bispectral index monitoring not used for sevoflurane? Could this influence anesthesia depth and hence outcome?

4. The authors state that the Pringle maneuver was used intermittently but do not include the duration of clamping and how many cycles were used between treatment groups. The Pringle maneuver has been shown to be protective (a form of ischemic pre-conditioning) so differences here could affect the outcomes measured. Were there any differences between the application of the Pringle maneuver?

5. Post-operative serious adverse events should be listed (supplemental would be fine) beyond their classification as grade 3a,4b etc. Not all readers will know what these various grades mean, so they should be included.

6. The authors present their data two ways, intention to treat (ITT) or “per-protocol” analysis. In the latter analysis, the authors drop 1-2 patients per treatment arm due to incomplete protocol implementation. The data/findings do not change, but the statistical data does change. In other words, it did not affect their interpretation, so why include it? The paper becomes much denser by the almost duplication of data without an effect on its interpretation. To make the paper more readable, I would suggest moving the “per-protocol” analysis to a supplemental figure.

7. The paper would benefit from English editing. Some sentences are non-sensical (“To reach AST, ALT and tBil peaks within 3 POD were confirmed by retrospective data analysis of 72 patients in liver resection for metastatic liver tumors at NCCH”).

6. PLOS authors have the option to publish the peer review history of their article (what does this mean?). If published, this will include your full peer review and any attached files.

Reviewer #1: No

Reviewer #2: **Yes: **Timothy Angelotti MD PhD

---

## [Author Response · Author response to Decision Letter 0]

7 Jul 2023

Journal Requirements:

Thank you for point to note.

1.Please ensure that your manuscript meets PLOS ONE's style requirements, including those for file naming. 

→We are sorry about this matter. We change our manuscript to meet PLOS ONE style.

2.Please upload a new copy of Figure 1 as the detail is not clear. 

→We are sorry about this matter. I uploaded a new copy of Figure 1.

3.Please include captions for your Supporting Information files at the end of your manuscript, and update any in-text citations to match accordingly. 

→We are sorry about this matter. We include captions for Supporting information files at the end of manuscript.

4.Please review your reference list to ensure that it is complete and correct. If you have cited papers that have been retracted, please include the rationale for doing so in the manuscript text, or remove these references and replace them with relevant current references. Any changes to the reference list should be mentioned in the rebuttal letter that accompanies your revised manuscript. If you need to cite a retracted article, indicate the article’s retracted status in the References list and also include a citation and full reference for the retraction notice.

→Thank you for pointing it out. We remove retracted paper and replace fixed paper.

Additional Editor Comments:

Thank you for your kindly check. 

1. Methodology: in perioperative management , titration of propofol and sevoflurane should be based both on the BIS value to have solid comparison based on np differences.

→Thank you for pointing it out. As propofol can not use direct biological monitoring in clinical setting, anesthesia by propofol need to be used other non-direct monitoring like BIS. In turn, as volatile anesthetic agents can use end-tidal concentration which is direct biological monitoring, anesthesia by volatile anesthetic agent like sevoflurane does not necessarily need BIS in clinical settings. Consequently, we did not use BIS monitor at sevoflurane anesthesia. We add this matter at limitation in Discussion.

2. in the manuscript the author have to clarify both ( Anesthesia time surgery time ) , it should be stated from exactly when to when.

→Thank you for pointing it out. We state from when to when about anesthesia time and surgery time at measurement in Materials and Methods. 

3. All tables should have a caption clearly stating the type of date and how is it expressed and the P value of statistical significance.

→Thank you for pointing it out. We state the type of data and how is it expressed and the P value of statistical significance at a caption in all tables

4. Figure 1 has neither title nor caption.

→Thank you for pointing it out. I wrote title and caption of Figure 1 at Patients in Results.

5. In the pre-protocol the author showed the operative time to be significantly longer in the Sevoflurane group,,, why ?

→Thank you for pointing it out. As according to the opinion from Reviewer#1, we canceled the statistical comparisons, the significant difference is dismissed. However as the operative time was long in the sevoflurane group, we address our consideration at Limitation in Discussion. As the difference between anesthesia time and operation time and total ischemic time which represented the duration of liver resection were similar, we assess the difference was mainly caused by the peering adhesion for previous surgery including primary cancer resection and did not largely affect the outcomes. 

6. Discussion: paragraph 1, in the last line , the author said (anesthetic agent was significantly affected with postoperative peak levels of transaminase.) what is meant by this phrase?

→Thank you for pointing it out. We changed this sentence and used English editing service.

Reviewer #1: 

1. I have few (minor/little) doubts regarding ‘Sample size estimation’. Firstly, you said “We set a difference of 300 IU/L in postoperative peak AST levels between the two groups” but the basis of this is not given. Though later as you specified since it is felt to be ‘clinically significant’ is perfectly alright but mind you that unless any reference is quoted, it is treated as ‘subjective’ [I am not a clinician, however, general opinion is expressed]. For standard deviation you said “was decided by referring to retrospective data of 52 patients” but the source of these data is not revealed. At the end of this section you said “this study was adequately powered with 28 patients per group (total 56 patients)” is doubtful because according to table-2 on page 158 of Jacob Cohen’s paper “A power primer” in Psychological Bulletin, 1992, vol.:112, pp 155-159 [which is a sort of summary of the excellent book by Cohen himself titled ‘Statistical power analysis for the behavioral sciences’, Academic Press, 1977, New York] even for medium effect size you need n=64 per group (type-I error=0.05, power=80%). Fortunately, the power achieved in this trial is not very low (75.69% by referring to table-2 of this manuscript), however, here the comparison pertains only to ‘post’ values and baseline is not given any recognition [I am not sure regarding its clinical importance/relevance]. Look at the absolute values [both mead S.D.] in table-2 {Peak AST level (IU/l) Sevoflurane group (n=28) 510.3 [362.4] Propofol group (n=28) }291.7 [151.2]}. Any software or manual calculations will show that your ‘sample size’ results are definitely correct [as an example, a ‘Screen-shot’ of output from software called COMPARE2 is pasted below – which shows that I have no doubt about correctness of calculations], I doubt only about assumptions [assumed values of a difference of 300 IU/L and a standard deviation of 400 IU/L]. This implies that a very large effect size is assumed and that is doubtful. Example: Screen-shot of output from COMPARE2

→Thank you for pointing it out and kindly teaching about statistical matter. We are sorry but we can not recruit more participants in this study. Instead, we add the rationale about difference setting at Sample size estimation in Materials and Methods and the preliminary data set as supplemental data. And we erase the word ‘ adequately’ and add the wide range of standard deviation in sample size estimation (in addition, the power achieved in this study) at Limitation in Discussion.

2.Note that the comparison of baseline characteristics when random allocation/assignment is used/done is not required [‘P’ values in last column 4 of Table 1 - Patients characteristics]. In this context, please read [though I am sure that the authors already know these things] a note which is pasted from one famous standard textbook on ‘Medical Research Methodology’:To provide a description of baseline characteristics is entirely reasonable (since it is clearly important in assessing to whom the results of the trial can be applied), however, statistical comparison of baseline characteristics when random allocation/assignment is used/done [often for good/standard/leading journals these days] is not required, because even if P-value(s) turn(s) out to be significant (while comparing baseline characteristics despite random allocation), it is, by definition, a false positive as you then are supposed to be testing ‘randomization’ then, which in any single trial may not balance all baseline characteristics (particularly when sample sizes are small). Remember that ‘randomization’ is a sort of ‘insurance’ and not a guarantee scheme. Authors may please refer to following articles:

References:

1. Stuart J. Pocock, et al., ‘Subgroup analysis, covariate adjustment and baseline comparisons in clinical trial reporting: current practice and problems’, Statistics in medicine, 2002; 21:2917–2930 [Particularly page 2927]

2. Harrington D, et al., ‘New guidelines for statistical reporting in the journal’, N Engl J Med 2019;381:285-6

[Important message (indirectly/ultimately indicated) from these articles: Never do any comparison with respect to ‘baseline’ characteristics {by applying statistical significance test(s)}, when allocation is done randomly].

However, Statistical comparison [only with respect to important/indicated variables] of baseline characteristics may be performed, to find out if analysis adjustment (say stratified analyses or else) is required with respect to these variables.

→Thank you for pointing it out and kindly teaching about statistical matter. We agree with your opinion. We erase statistical comparison.

3. In ‘Abstract- Results’ section [and also in ‘outcome - Per-protocol analysis’ section later] there is a sentence “At multiple linear regression …..” which, in my opinion, is not grammatically correct {instead of At, it should be As shown by} . Please check. Later in ‘Randomization’ section, a sentence “A computer generated the blocked randomization with no stratification” appears to be incomplete. In section ‘Statistical analyses’, instead of “We expressed the assessed variables….” it could/should [desired] be “We summarised the assessed variables….”, I think.

→Thank you for pointing it out. We correct these points and used English editing service for checking our English. 

4. Highlighting (listing/pointing out) the “Limitations” {with possible effects} is highly appreciable (although lower sample size is not included starting ‘Discussion’ with “In this adequately powered RCT is not agreed). However, mind you that as pointed out in ‘important note’ above “This review pertains only to ‘statistical aspects’ of the study and so ‘clinical aspects’ should be assessed separately/independently [preferably by an expert surgeon in concerned field]. Some of these limitations may have significant influence on results.

→Thank you for pointing it out. We correct these points and use English editing service.. 

Reviewer #2:

1. The Introduction discusses prior research in this area but should also include a better description of the differences in protocols between the prior study (See Discussion Page 22 “The patients were mainly anesthetized in…. usual clinical methods”). By including this discussion in the Introduction would better inform the reader about why this study is important. The authors used a protocol that more closely mimicked traditional anesthetic practice compared to prior work by others. This point should be emphasized more.

→Thank you for your suggestion. We add the sentence to emphasize our strength at Introduction.

2. The authors excluded patients with liver cirrhosis, only included patients with similar tumor burdens etc. to produce a more homogenous patient population (see last paragraph in the Discussion). Such details should be emphasized in the introduction to build interest in the reader.

→Thank you for your suggestion. We add the sentence to emphasize our strength at Introduction.

3. Propofol was titrated based upon bispectral index monitoring, but sevoflurane was titrated to end-tidal concentration. Why was bispectral index monitoring not used for sevoflurane? Could this influence anesthesia depth and hence outcome?

→Thank you for pointing it out. As propofol can not use direct biological monitoring in clinical setting, anesthesia by propofol need to be used other non-direct monitoring like BIS. In turn, as volatile anesthetic agents can use end-tidal concentration which is direct biological monitoring, anesthesia by volatile anesthetic agent like sevoflurane does not necessarily need BIS in clinical settings. Consequently, we did not use BIS monitor at sevoflurane anesthesia. We assess this does not influence anesthesia depth and outcome. We add this matter at limitation.

4. The authors state that the Pringle maneuver was used intermittently but do not include the duration of clamping and how many cycles were used between treatment groups. The Pringle maneuver has been shown to be protective (a form of ischemic pre-conditioning) so differences here could affect the outcomes measured. Were there any differences between the application of the Pringle maneuver?

→Thank you for pointing it out. We assessed the duration of clamping as total ischemic time (TIT) and add the number of the cycle of Pringle maneuver at Measurement in Materials and Methods. Consequently, we assess the application of the Pringle maneuver is no difference between groups.

5. Post-operative serious adverse events should be listed (supplemental would be fine) beyond their classification as grade 3a,4b etc. Not all readers will know what these various grades mean, so they should be included.

→Thank you for pointing it out. We add the details of serious adverse event at Outcome in Results.

6. The authors present their data two ways, intention to treat (ITT) or “per-protocol” analysis. In the latter analysis, the authors drop 1-2 patients per treatment arm due to incomplete protocol implementation. The data/findings do not change, but the statistical data does change. In other words, it did not affect their interpretation, so why include it? The paper becomes much denser by the almost duplication of data without an effect on its interpretation. To make the paper more readable, I would suggest moving the “per-protocol” analysis to a supplemental figure.

→Thank you for your suggesting. We moved the per- protocol analysis to supplement files.

7. The paper would benefit from English editing. Some sentences are non-sensical (“To reach AST, ALT and tBil peaks within 3 POD were confirmed by retrospective data analysis of 72 patients in liver resection for metastatic liver tumors at NCCH”).

→Thank you for your suggestion. We use English editing service.

---

## [Editor Report · Decision Letter 1]

4 Aug 2023

Protective effect of propofol compared with sevoflurane on liver function after hepatectomy with Pringle maneuver: a randomized clinical trial

PONE-D-23-10418R1

Dear Dr. Matsumi,

We’re pleased to inform you that your manuscript has been judged scientifically suitable for publication and will be formally accepted for publication once it meets all outstanding technical requirements.

Kind regards,

Hossam Eldien Ahmed Anis ElShamaa, M.D.

Academic Editor

PLOS ONE
---

## [Editor Report · Acceptance letter]

16 Aug 2023

PONE-D-23-10418R1 

Protective effect of propofol compared with sevoflurane on liver function after hepatectomy with Pringle maneuver: a randomized clinical trial 

Dear Dr. Matsumi:

I'm pleased to inform you that your manuscript has been deemed suitable for publication in PLOS ONE. Congratulations! Your manuscript is now with our production department. 

Kind regards, 

on behalf of

Dr. Hossam Eldien Ahmed Anis ElShamaa 

Academic Editor

PLOS ONE